



# Magnetotail Reconnection Asymmetries in a Small, Earth-Like Magnetosphere

Christopher M. Bard[1] and John C. Dorelli[1]

[1]Geospace Physics Lab, NASA Goddard Space Flight Center

**Correspondence:** Chris Bard (christopher.bard@nasa.gov)

**Abstract.** We use a newly developed global Hall MHD code to investigate how reconnection drives magnetotail asymmetries in small magnetospheres. Here, we consider a scaled-down, Earth-like magnetosphere where the ion inertial length ($\delta_i$) is artificially inflated to one planetary radius (the real Earth's $\delta_i \approx 1/15 - 1/20\ R_E$ in the magnetotail). This results in a magnetotail width on the order of $30\ \delta_i$, slightly smaller than Mercury's tail and much smaller than Earth's. At this small size, we find that the Hall effect has significant impact on the global flow pattern, changing from a symmetric, Dungey-like convection under resistive MHD to an asymmetric pattern similar to that found in previous Hall MHD simulations of Ganymede's subsonic magnetosphere as well as other simulations of Mercury's using multi-fluid or embedded kinetic physics. We demonstrate that the Hall effect is sufficient to induce a dawnward asymmetry in observed dipolarization front locations and find quasi-periodic global scale dipolarizations under steady, southward solar wind conditions. On average, we find a thinner current sheet dawnward; however, the measured thickness oscillates with the dipolarization cycle. During the flux-pileup stage, the dawnward current sheet can be thicker than the duskward sheet. This could be an explanation for recent observations that suggest Mercury's current sheet is actually thicker on the duskside: a sampling bias due to a longer-lasting "thick" state in the sheet.

## 1 Introduction

In the magnetospheres of Mercury and Earth, observations of plasmoids, flux bundles, and dipolarization fronts (DFs) demonstrate a marked asymmetry in their distribution across the magnetotail. At Earth, a number of studies have found magnetotail duskward biases in several magnetic phenomena: flux rope occurrence (Slavin et al., 2005; Imber et al., 2011), dipolarization fronts (Liu et al., 2013), energetic particle injections (Gabrielse et al., 2014), and reconnection (e.g. Asano et al. (2004); Genestreti et al. (2014)). Additionally, the current sheet was found to be thinner on the duskside (Artemyev et al., 2011; Vasko et al., 2015). Similarly, at Mercury, Poh et al. (2017b) used MESSENGER data to fit the Harris sheet model to 234 tail current sheet crossings and found a bias towards dusk having thinner current sheets (by $\approx 10 - 30\%$). In contrast, however, other





MESSENGER studies (Sun et al., 2016; Dewey et al., 2018) found dawnward biases in dipolarization events and reconnection front locations.

The general existence of tail asymmetry is thought to be a result of sub-ion-scale effects (Lu et al., 2018; Liu et al., 2019), though there is still some uncertainty about the exact manifestation and causes of specific asymmetries. It is debated whether Hall electric fields are sufficient to reproduce this or if other ion/electron scale scale physics are required. Although some authors argue that electron-scale physics is required (Chen et al., 2019), we show in this paper that Hall effects are sufficient to cause an asymmetry in some observed features. Furthermore, it is unknown exactly why Mercury and Earth observe dif-

ferent asymmetries; it is hypothesized that system size effects (relative to the ion inertial length $\delta_i$) play a key role (Lu et al., 2016, 2018; Liu et al., 2019).

Several studies have proposed mechanisms to explain how Hall reconnection induces asymmetry in the magnetotail. Lu et al. (2016, 2018) (hereafter: Lu+), in studying Earth's magnetotail with global hybrid simulations and localized PIC simulations, showed that the decoupling of ions and electrons within the current sheet (the Hall effect; e.g. Sonnerup (1979)) creates a

electric field and associated tail current density. The resulting $\boldsymbol{E} \times \boldsymbol{B}$ drift is sufficient to create tail asymmetries and indeed may be the primary cause. The duskside magnetic flux is preferentially evacuated via electron transport dawnward, which leads to a smaller normal $B_z$ and thinner current sheet on the duskside.

In a similar study, Liu et al. (2019) (hereafter: Liu+), using local PIC simulations of embedded, thin current sheets, confirmed that the Hall effect creates electron $\boldsymbol{E} \times \boldsymbol{B}$ and diamagnetic drifts which transport magnetic flux dawnward within the current

sheet. However, they found that, although the pre-existing tail $B_z$ initially suppresses the onset of dawnside reconnection, the reconnection $B_z$ drives outflows towards dawn and thins out the current sheet on that side. This creates an "active region" of reconnection on the dawn side, which has a thinner current sheet and stronger tail current $j_y$. After analyzing both these studies, Liu+ proposed that, although the Lu+ model provides a explanation for a duskward bias in the *initial* reconnection onset, the Liu+"active region" provides an explanation for dawnward biases within local, in-progress magnetotail reconnection. We test

several aspects of this general picture within this paper.

Unfortunately, simulating large magnetospheres such as Earth (few hundred $\delta_i$) while properly resolving the small-scale Hall physics requires grid sizes in the billions of cells. Several strategies have been proposed to evade this constraint; one is to embed regions of detailed kinetic physics within large-scale ideal MHD simulations (Chen et al., 2019). This allows for reproduction of kinetic effects within certain regions of the magnetosphere without having to run an expensive, fully kinetic

simulation. However, these simulations assume no kinetic effects outside the embedded regions, which are limited to certain regions in the dayside and/or the tail.

Another strategy suggests that we need only set the Hall scale to some length sufficient to capture the essential physics of Hall reconnection without having to fully resolve the physical length scale. In these simulations, the Hall length is set to $\approx 3\%$ of the global scale length (Tóth et al., 2017) which is sufficient to capture the out-of-plane flows and the quadrupolar

magnetic field structure induced by the Hall effect. However, recent research in 2D island coalescence (Bard and Dorelli, 2018) suggests that although including the Hall term in MHD simulations is sufficient in itself to generate these signatures of Hall reconnection, the actual reconnection rate depends on resolution and numerical resistivity. Although the Hall term is





present, the reconnection itself may be Sweet-Parker-like and slow (unlike fast Hall reconnection). Bard and Dorelli (2018) observed that 20-25 cells per $\delta_i$ was necessary (within the context of their numerical viscosity) in order to observe fast Hall
reconnection. This is much greater than the $5-10$ cells per ion inertial length typically used in simulations (Dorelli et al., 2015; Dong et al., 2019; Chen et al., 2019). This suggests that, although artificially inflating $\delta_i$ allows the Hall effect to emerge and have a global impact, much higher resolution is required to observe the universally fast ($\sim 0.1\ v_A$) reconnection observed in kinetic simulations. Finally, Bard and Dorelli (2018) found qualitatively different behavior for varying ratios of system size to $\delta_i$: large systems can produce bursty reconnection (with a low average reconnection rate) even when $\delta_i$ is sufficiently resolved
to produce "fast", instantaneous reconnection. Ultimately, these effects mean that much higher resolution than is currently attainable will be needed to properly model global systems.

These models will require enormous computing power. Over the last decade, graphics processing units (GPUs) have proven to be a robust and viable basis for scientific computing. Indeed, several groups have already utilized GPUs to accelerate plasma simulations throughout heliophysics, astrophsyics, and plasma physics (Bard and Dorelli, 2014; Benítez-Llambay and Masset,
2016; Fatemi et al., 2017; Bard and Dorelli, 2018; Schive et al., 2018; Grete et al., 2019; Liska et al., 2019; Wang et al., 2019).

In this paper, we undertake a numerical experiment designed to assess the the role of the Hall effect on global magnetospheric structure and dynamics within a "small" magnetosphere, specifically focusing on how it induces asymmetry in the magnetotail. We present a magnetosphere simulation code which accelerates the explicit MHD solver algorithm via GPUs. We simulate an Earth-like analogue magnetosphere which has a similar bow shock-magnetopause distance and magnetotail width as Earth's
(relative to the planetary radius); however, the ion inertial length is artificially inflated to the planetary radius ($\delta_i = R_p$). This magnetosphere is "small" relative to the ion inertial length (magnetotail width $\approx 30 d_i$; Earth's magnetotail $\approx 600 d_i$), meaning that Hall physics plays a greater role in magnetospheric dynamics and that global effects are more readily observed. We view this work as a first step in the study of the system-size dependence of magnetic reconnection in Earth-like magnetospheres; future system-size studies can be performed by making $\delta_i$ smaller relative to the planetary radius and increasing the resolution
to sufficiently cover the ion scales.

This paper is presented as follows: Section 2 provides a brief overview of the Hall MHD algorithm as implemented using GPUs; Section 3 provides the initial condition and setup of the simulation; Section 4 presents tail asymmetries in the simulation and discusses them in the context of observations and proposed theoretical explanations.

## 2   Methods and Code

We take a Hall MHD code accelerated by graphics processing units using the MPI and NVIDIA CUDA libraries (Bard and Dorelli, 2014; Bard, 2016; Bard and Dorelli, 2018) and adapt it to simulate planetary magnetospheres. We review the underlying mathematical equations and algorithms in this section.

Following Powell et al. (1999), we split the magnetic field vector $\boldsymbol{B}$ into a background component $\boldsymbol{B}_g$ and a perturbed, evolving component $\boldsymbol{B}_1$ such that $\boldsymbol{B} = \boldsymbol{B}_1 + \boldsymbol{B}_g$. The embedded $\boldsymbol{B}_g$ is assumed to be static ($\partial \boldsymbol{B}_g / \partial t = 0$), divergence-free
($\nabla \cdot \boldsymbol{B}_g = 0$), and curl-free ($\nabla \times \boldsymbol{B}_g = 0$). This allows for more accurate handling of the magnetic field, especially near the





planet where the dipole field is very strong. In order to preserve the divergence-free constraint on the evolved magnetic field, we solve the "Generalized Lagrangian Multiplier" (GLM) formulation of MHD (Dedner et al., 2002), with an additional Hall term added via Ohm's Law.

The ideal MHD Ohm's law is extended with the Hall term such that the electric field $\boldsymbol{E}$ is given by

$$\boldsymbol{E} = -\frac{\boldsymbol{v} \times \boldsymbol{B}}{c} - \frac{\boldsymbol{J} \times \boldsymbol{B}}{nec} , \tag{1}$$

with $c$ the speed of light, $e$ the electron charge, $n$ the plasma number density, $\boldsymbol{v}$ the plasma bulk velocity vector, and $\boldsymbol{J} = \frac{c}{4\pi}\nabla \times \boldsymbol{B}$ the current density vector. We note that since the background magnetic field $\boldsymbol{B_g}$ is curl-free in our formulation, the current density is taken to be the curl of the perturbation $\boldsymbol{B_1}$.

We normalize the density ($\rho$), magnetic field, and length scale to reference values $\rho_0$, $B_0$, and $L_0$, respectively. $\boldsymbol{v}$ is normalized to $v_0 = v_A = B_0/\sqrt{4\pi\rho_0}$, the pressure $P$ to $P_0 = B_0^2/(4\pi)$, and the time $t$ to $t_0 = L_0/v_0$. This results in the set of equations:

$$\frac{\partial \rho}{\partial t} + \nabla \cdot (\rho \mathbf{v}) = 0, \tag{2}$$

$$\frac{\partial \rho \mathbf{v}}{\partial t} + \nabla \cdot \left[ \rho \mathbf{v}\mathbf{v} + (p + \frac{B_1^2}{2} + \mathbf{B}_g \cdot \mathbf{B}_1)\mathbb{I} - \mathbf{B}_1\mathbf{B}_1 - \mathbf{B}_g\mathbf{B}_1 - \mathbf{B}_1\mathbf{B}_g \right] = 0, \tag{3}$$

$$\frac{\partial \mathcal{E}}{\partial t} + \nabla \cdot \left[ (\frac{\rho v^2}{2} + \frac{\gamma}{\gamma - 1}p)\mathbf{v} + B_1^2\mathbf{v}_T + (\mathbf{B}_g \cdot \mathbf{B}_1)\mathbf{v}_T - (\mathbf{v}_T \cdot \mathbf{B}_1)(\mathbf{B}_g + \mathbf{B}_1) \right] = 0, \tag{4}$$

$$\frac{\partial \mathbf{B}_1}{\partial t} + \nabla \cdot [\mathbf{v}_T\mathbf{B} - \mathbf{B}\mathbf{v}_T] + \nabla\psi = 0, \tag{5}$$

$$\frac{\partial \psi}{\partial t} + c_h^2 \nabla \cdot \mathbf{B} = -\frac{c_h^2}{c_p^2}\psi, \tag{6}$$

where $\mathcal{E} = \rho v^2/2 + p/(\gamma - 1) + B_1^2/2$ is the total energy density, $\gamma$ is the ratio of specific heats (taken to be 5/3 in all of our simulations), and $\mathbf{v}_T = \mathbf{v} + \mathbf{v}_H$ combines the bulk velocity ($\mathbf{v}$) with the normalized Hall velocity $\mathbf{v}_H = -\bar{\delta}_i\mathbf{J}/\rho$. The ion inertial length $\delta_i = c\sqrt{m_i}/\sqrt{4\pi n_0 e^2}$ is normalized to the reference length such that $\bar{\delta}_i = \delta_i/L_0$. The normalized $\bar{\delta}_i$ in our simulation is an user-set fixed parameter that can be changed at run-time. We evaluate the normalized current density ($\mathbf{J} = \nabla \times \boldsymbol{B_1}$) at cell centers and linearly interpolate to the cell edges when needed. The resulting form is nearly identical to the second-order algorithm with averaging and central differences used to calculate $\boldsymbol{J}$ in Tóth et al. (2008).

$\psi$ is a scalar function whose evolution is designed to be equivalent to $\nabla \cdot \mathbf{B}$; $c_h$ and $c_p$ are parameters for the propagation and dissipation of local $\mathbf{B}$ divergence errors, respectively. Following Dedner et al. (2002), we set $c_h$ as the global maximum wave speed over the individual cells and set $c_p$ such that $c_p^2/c_h$ is within the range $0.05 - 0.5$. Although Dedner et al. (2002) recommended $c_p^2/c_h = 0.18$ and this value works very well to control the magnetic divergence in non-magnetospheric simulations, we find that some level of tweaking is required because of the accumulation of divergence errors at the inner boundary. To ameliorate complications caused by this issue, we separate the momentum equation into a non-magnetic flux and a magnetic





source term:

$$\frac{\partial \rho \boldsymbol{v}}{\partial t} + \nabla \cdot [\rho \boldsymbol{v}\boldsymbol{v} + p\mathbb{I}] = \boldsymbol{J} \times (\boldsymbol{B_1} + \boldsymbol{B_g}) \; , \tag{7}$$

which prevents divergence errors from inducing a non-physical acceleration along magnetic field lines (Brackbill and Barnes, 1980), but with some loss of accuracy in evaluating the momentum evolution.

We note that constrained transport methods (e.g. Evans and Hawley (1988); Balsara and Spicer (1999); Dai and Woodward (1998); Londrillo and del Zanna (2004); Stone et al. (2008); Lee and Deane (2009)) are a way to evaluate the magnetic field such that divergence is enforced to machine precision. However, we have found the divergence cleaning+source term method simpler to implement, especially with regards to the magnetosphere-planetary boundary interface.

The overall system is evolved via a time-explicit second-order Runge-Kutta scheme coupled with a simple HLL Riemann solver (Harten et al., 1983; Toro, 1999) and a monotonized central limiter (e.g. Tóth et al. (2008)) with the slope-limiting parameter $\beta$ set to $1.25$.

## 3  Problem Initialization

For our mini-Earth, we choose normalized solar wind and terrestrial magnetic field parameters such that its magnetopause standoff distance (in planetary radii) matches that of Earth's magnetosphere ($\approx 8 - 10 R_p$) and that the ion inertial length is equivalent to the planetary radius. The "mini" aspect to our magnetosphere comes from the relative importance of ion scale physics: we set the unit length $L_0 = R_p$ and set $\delta_i/L_0$ to unity (compare to at Earth: $\delta_i/R_p \approx 1/60, 1/70$).

The solar wind parameters are: $\rho_{sw} = 1 \; \rho_0$, $v_{sw} = 4.09 \; v_0$, and wind plasma $\beta_{sw} = 0.305$ such that $P_{sw} = 0.1526 \; P_0$. The wind magnetic field is initially set to $\boldsymbol{B_{sw}} = (-0.174, 0, 0.985) B_0$ for a northward IMF with magnitude $B_{sw} = 1 \; B_0$; we later flip the IMF by setting $Bz = -0.985$.

The planetary background magnetic field ($\boldsymbol{B_g}$) is approximated with a magnetic dipole

$$\boldsymbol{B_g} = \frac{3(\boldsymbol{m} \cdot \boldsymbol{r})\boldsymbol{r} - \boldsymbol{m}\|r\|^2}{\|r\|^5} \tag{8}$$

with $\boldsymbol{r}$ the position vector from the center at $(0,0)$ and the planetary dipole moment $\boldsymbol{m} = (M_x, M_y, M_z)$ taken as $(0, 0, -3000)$, such that $\|\boldsymbol{B_g}\| = 3000 \; B_0$. This satisfies the requirement that $\boldsymbol{B_g}$ be both curl-free and divergence-free.

We tried various prescriptions for the inner boundary, including floating (zero-gradient) and fixing various combinations of different variables. Specifically, we had issues with density depletion at the boundary which resulted in large local Alfvén speeds and small global timesteps. Ultimately, although the following inner boundary conditions are not entirely realistic, they allow a stable evolution of the magnetosphere in both the dayside and the tail. We set the inner boundary at a radius of $3 \; R_P$; in these ghost cells, we fix the density at $4 \; \rho_0$, float the pressure, float the radial magnetic field, set the tangential **B** to zero, and set the velocity to zero. For the divergence cleaning, we find that simply setting the ghost $\psi_c = 0$ works better than having a floating condition. We note that in more realistic magnetospheres, cold plasma from the ionosphere may flow out to the tail and impact the dynamics. We will leave this topic to future studies.





150    For the outer boundaries, the left edge of the simulation domain fixes the conservative variables to the background solar wind condition; the rest of the box has zero-derivative boundaries for all variables.

The simulation coordinates are defined with $-X$ pointing towards the Sun, $Z$ along the planetary magnetic dipole axis, and $+Y$ towards the dusk completing the orthogonal set. In order to resolve the artificially inflated ion inertial length, we choose 5 cells per $\bar{\bar{\delta}}_i$, giving a minimum resolution of $\Delta x, \Delta y, \Delta z = 0.2L_0$. This resolution is set within the range $-20\,L_0 < x < 20\,L_0; -15\,L_0 < y, z < 15\,L_0$; beyond this the cell length increases by 7% with each additional cell up to a maximum of $5R_E$ or until it hits the boundary. The total size of the grid is $290 \times 253 \times 253$, or just over 18 million cells.

Typically, 10 cells is used to resolve $\delta_i$. However, previous results with island coalescence (Bard and Dorelli, 2018) suggest that 5 cell resolution is sufficient for our code to obtain signatures of Hall reconnection, namely the quadrupolar magnetic field structure and the related out-of-plane reconnection outflow. In either case, however, the reconnection is still slow and Sweet-Parker-like; Bard and Dorelli (2018) suggest that 20-25 cells/$\delta_i$ is required to recover the fast Hall reconnection found by, e.g. Shay et al. (2001). Thus, the difference between 5 and 10 cells/$\delta_i$ is not significant enough to run the higher-resolution, more computationally expensive simulation, especially for our goal of assessing the global impact of Hall physics on the magnetosphere. We do note that, because $\delta_i \propto 1/\sqrt{\rho}$, a higher resolution does provide more of a buffer against the variability of local $\delta_i$ due to density fluctuations.

165    We start the simulation in ideal MHD ($\bar{\bar{\delta}}_i = 0$) with a northward IMF ($\boldsymbol{B}_{sw}$ given above) for $120\,t_0$, and then flip $B_{z,sw}$ for the southward IMF case and run it for another $120\,t_0$. At this point, we turn on the Hall term by setting $\bar{\bar{\delta}}_i = 1$ and run it for another $12\,t_0$ in order to allow the perturbations induced from the abrupt change of physics to settle. From this point on, the simulation was run for $45\,t_0$ under continuous pure southward IMF and with the Hall term on. The results discussed below all come from this portion of the run with the Hall term enabled.

## 4   Results and Discussion

### 4.1   Hall-induced asymmetry

Prior to turning on the Hall term, the magnetospheric convection is of Dungey-type. Turning the Hall term on, however, induces an out-of-reconnection-plane $\boldsymbol{E} \times \boldsymbol{B}$ force which breaks that symmetry and drives convection in a preferred direction (Figure 1). For smaller magnetospheres, this effect was first seen in non-ideal MHD simulations of Ganymede (Dorelli et al., 2015; Tóth et al., 2016; Wang et al., 2018); this was later seen in 10-moment and embedded-kinetic simulations of Mercury (Dong et al., 2019; Chen et al., 2019). Our simulation supports the idea that it is this Hall-induced drift which produces asymmetries; no kinetic effects are required.

### 4.2   Dipolarizations

In our simulation, the Hall electric field induced by tail reconnection accelerates ions towards the duskside and the electrons towards dawn. Since $\delta_i = R_E$ here, the reconnection current sheet spans a significant fraction across the tail; this means that

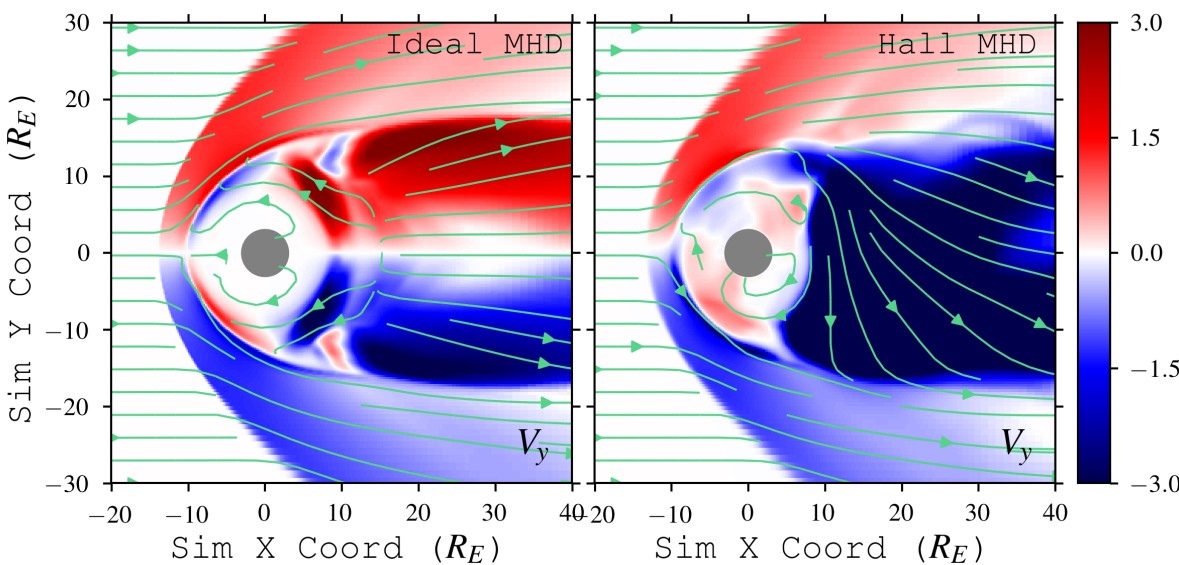

**Figure 1.** Cross-tail velocity $V_y$ in the tail plane perpendicular to reconnection for both ideal (left) and Hall (right) MHD, normalized to $v_0$. Streamlines show in-plane velocity. A typical Dungey-like, symmetric convection pattern induced by numerical resistivity is clearly demonstrated in ideal MHD. Adding the Hall term induces out-of-reconnection-plane flows which drives an asymmetric convection pattern; this is similar to what has been simulated for Ganymede (Dorelli et al., 2015; Tóth et al., 2016; Wang et al., 2018).

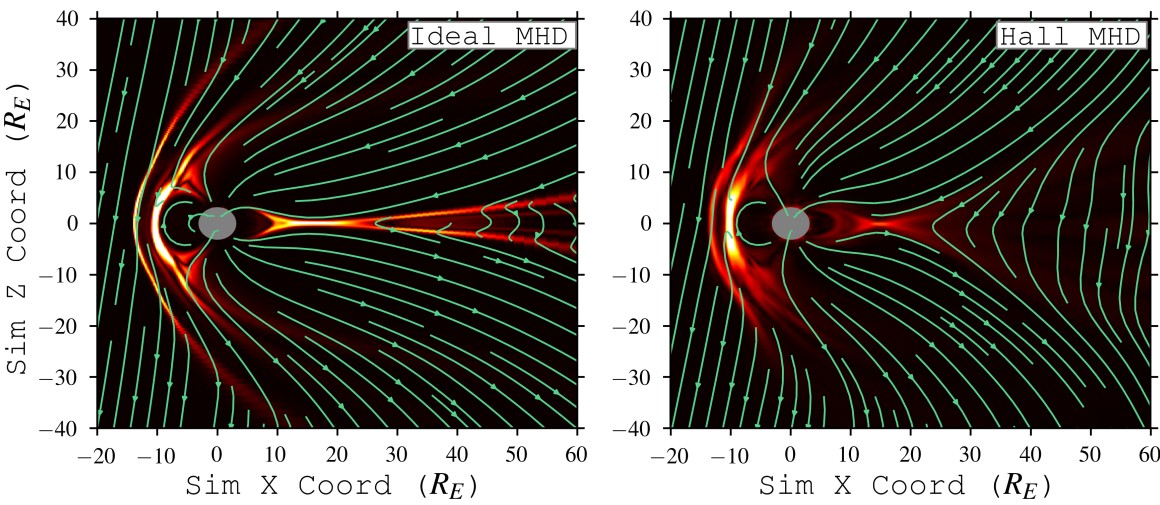

**Figure 2.** Normalized Current Density Magnitude $\|J\|$ in the simulation $xz$-plane at $y = 0$ for both ideal (left) and Hall (right) MHD. Streamlines show in-plane magnetic field. Adding the Hall term induces out-of-reconnection-plane flows (Fig. 1); the resulting tail convection causes the magnetotail current sheet to vary in width across the tail (see, e.g., Figs. 3 and 7 for examples of $\|J\|$ in the $xy$-plane).



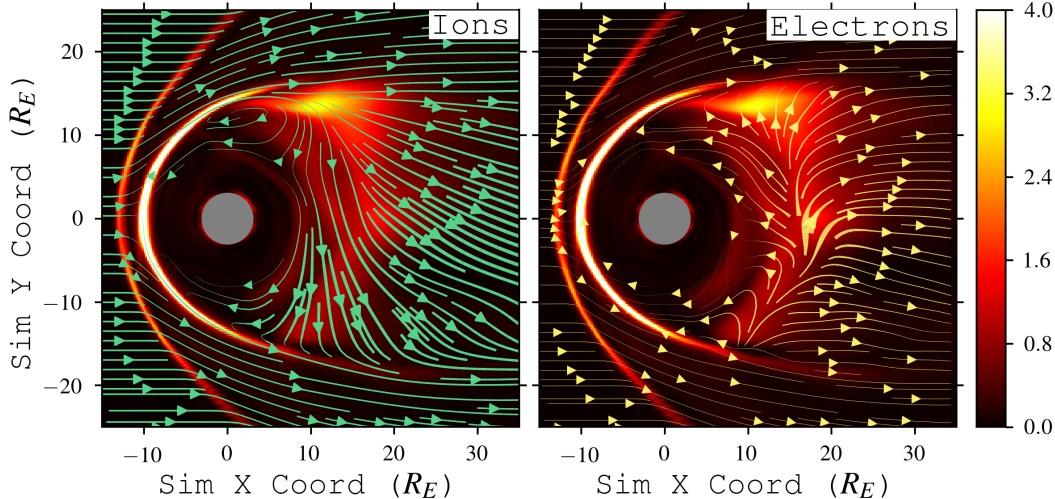

**Figure 3.** Cross-tail convection in the reconnection plane at $z = 0$, with background color (orange) illustrating relative current sheet density magnitude. Yellow (green) streamlines indicate direction of electron (ion) velocity, with line size proportional to magnitude and normalized relative to the maximum electron (ion) velocity in the plane. The electron velocity (in normalized units) is calculated from $\mathbf{v}_e = \mathbf{v} - \mathbf{J}/\rho$.

the ions are decoupled from the magnetic field during much of their in-plane convection duskward (blue arrows in Figure 3. The electrons, being coupled to the magnetic field, carry the reconnected, normal $B_z$ flux dawnward (yellow arrows). Because the reconnected magnetic flux originates over a large region within the tail, there is a significant pileup leading to a reconnecting, active region of plasmoid formation on the dawnside. This pileup + reconnection mechanism may be a general cause of dipolarizations in small magnetospheres (like Mercury, e.g. Sundberg et al. (2012)).

During the $45\,t_0$ duration of our simulation there were 7 events on the dawnside (none on the duskside) which followed the general substorm pattern of a buildup/loading phase followed by a unloading (or expansion/relaxation) phase (Rostoker et al., 1980). For each event, we observed pileup of the normal $B_z$ magnetic flux over a period of several $t_0$, followed by a burst of reconnection and the subsequent ejection of plasmoids tailward (Figs. 4 and 5). Three of the eight events produced large plasmoids (on the order of $10R_p = 10\delta_i$), while the rest resulted in smaller ones ($\leq 5R_p$; $\leq 5\delta_i$). The larger ejecta appeared to build up and release on timescales around $10\,t_0$, while the smaller events had shorter time scales around $5\,t_0$. Most events originated at a down-tail distance $\approx 13 - 16\,R_p$; after ejection, their resulting plasmoids traveled to about $30R_p$ down-tail over several $t_0$ before dissipating.

The observed dawnward bias in dipolarization events for our small magnetosphere corroborates similar dawnward biases found in MESSENGER observations (Sun et al., 2016; Dewey et al., 2018) and global simulations of Mercury (Dong et al., 2019; Chen et al., 2019). It is interesting to note that our results are under a steady, southward solar wind condition; continuous shifts between northward and southward IMF are not required to sustain generation of global substorms. As long as there is Hall-driven convection in the tail, the competition between dawnside $B_z$ pileup and reconnection will drive this cycle. At the



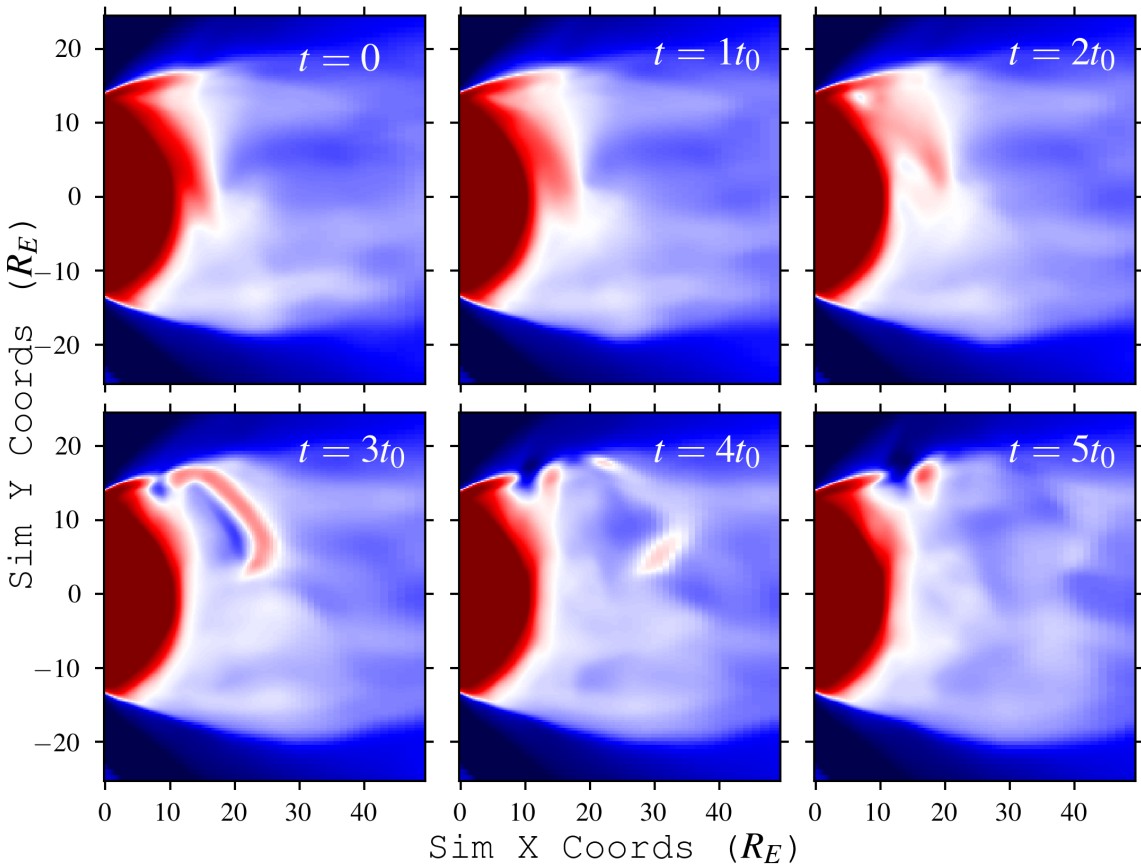

**Figure 4.** Formation and evolution of a global dipolarization over $5\ t_0$, as seen in the evolution of magnetotail normal magnetic field $B_z$ (red: out-of-page; blue: into page). Displayed times are relative to upper left image.

moment, it is not clear whether this process is unique to our mini-Earth, since its strong planetary dipole field means that
flux piles up over a large swath of the tail. It is possible that a similar process may occur at Mercury, i.e. that its observed
dipolarizations are indeed akin to global substorms (Kepko et al., 2015).

   We note that, at Earth, there are additional localized (i.e. not global) dipolarization fronts resulting from current sheet
instabilities or transient reconnection events (e.g. Runov et al. (2009); Sitnov et al. (2009)). We do not see these small-scale
fronts in our "mini-Earth"; this may be because we do not have enough down-tail resolution to observe localized current sheet
instabilities which form them.



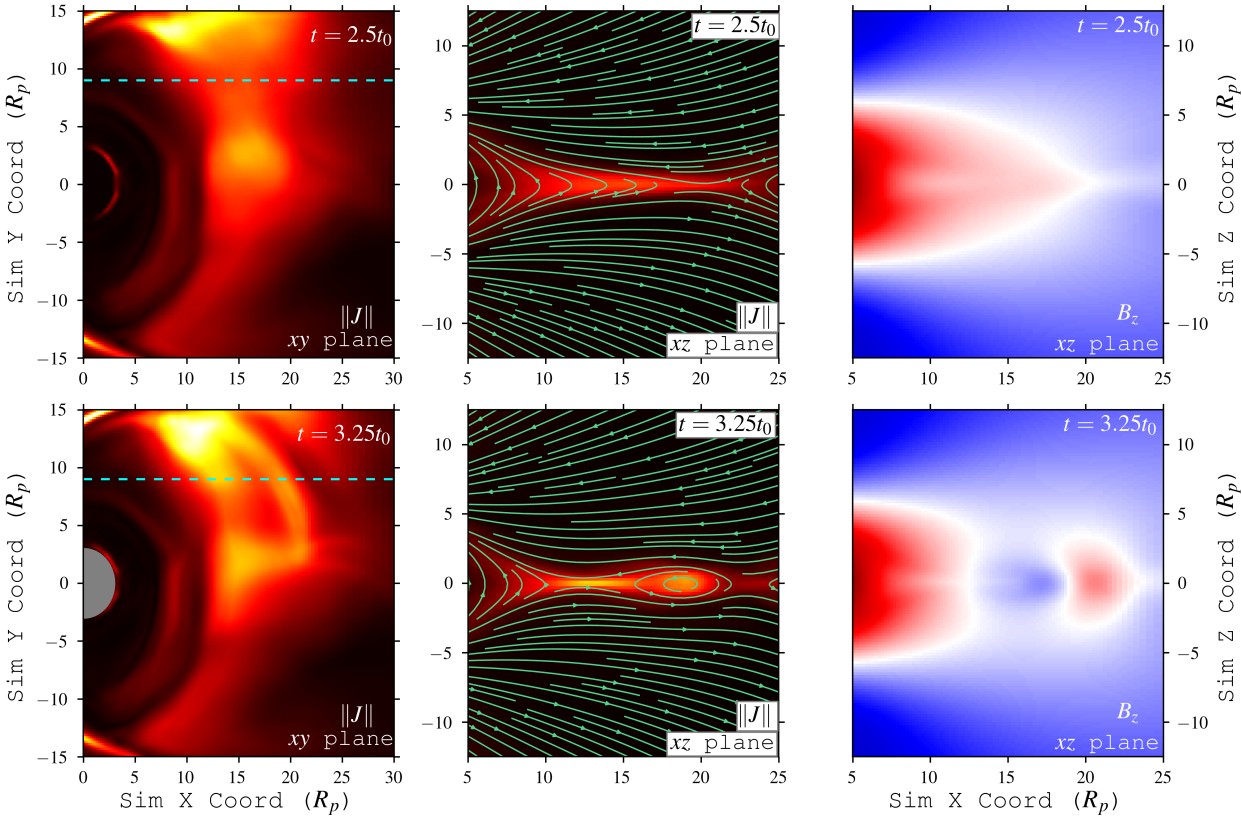

**Figure 5.** Formation and evolution of a global dipolarization due to locally reconnecting magnetic field lines in the $xy$ and $xz$ planes, as seen in the evolution of magnetotail current density ($\|J\|$) and normal magnetic field $B_z$. Displayed times are relative to the sequence shown in Fig. 4. The cyan line in the left figures mark the $y$-position of the $xz$ cuts in the center and right figures.

### 4.3 Current Sheet Thickness

Another test of the "active region" picture is the predicted thickness asymmetry of the tail current sheet: Liu+ predicted that the sheet would be thinner on the dawnside. We follow Poh et al. (2017a) and estimate the current sheet thickness in our model by using a Harris sheet (Harris, 1962):

$$B_x = B_0 \tanh \frac{Z - Z_0}{L_{CS}} + \text{offset} , \tag{9}$$

where $B_0$ is the asymptotic lobe field, $Z_0$ is the current sheet center, $L_{CS}$ is the current sheet half-thickness, and the offset allows for asymmetry between the north and south $B_x$ lobes on either side of the current sheet. We take 6000 one-dimensional cuts of $B_x$ along the north-south direction between $z = \pm 10\ R_E$ in a volume covering the current sheet from $12\ R_E < x < 16\ R_E$ and $-15\ R_E < y < 15\ R_E$, randomly sampled across the box plane and all times (example shown in Fig 6). These cuts are fit to eq. 9 using the Levenberg-Marquadt least-squares algorithm in scipy.curvefit (Virtanen et al., 2020); instances



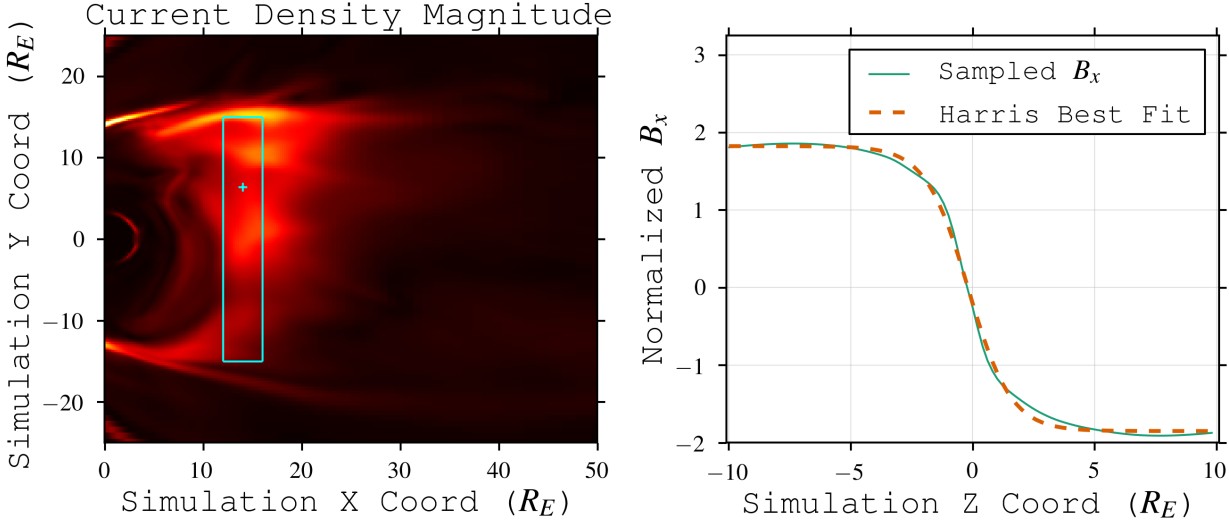

**Figure 6.** Example of $B_x$ sampling and Harris sheet fit (right figure) as described in text (eq. 9). The left figure shows the magnetotail current sheet magnitude in the simulation $z = 0$ plane; the $B_x$ sampling box domain boundaries are shown in cyan, with the small cross showing the location of the example sample. The box boundaries are $12 < x < 16$ and $-15 < y < 15$.

that do not fit well ($\chi^2 > 0.01$) or that return nonsensical results ($L_{CS} < 0$) are rejected. This results in 5037 samples of the current sheet thickness across the magnetotail (Figure 7). This distribution shows that the dawnward current sheet is thinner on average than the duskward sheet. However, there is a significant scatter in this result; the dawn sheet covers a wider range of thicknesses. This variation is caused by the dawnside pileup+reconnection mechanism.

The current sheet oscillates with the dipolarization cycle (Sec. 4.2) between a "thick state" due to the $B_z$ pileup and a "thin state" immediately following the flux unloading and plasmoid ejection. This is demonstrated in Fig. 8, where fitted CS thicknesses during both flux loading and unloading stages are plotted along with snapshots of the $B_z$ state. During the loading stage, the piled-up flux on the dawnside ($5R_E < y < 12R_E$) fattens the current sheet; here, the sampled dawn thicknesses are comparable to and can exceed the dusk thicknesses. However, after the unloading stage, the current sheet on the dawnside is much thinner where the flux has been evacuated (bottom right plot; $R > 15R_E$). Interestingly, we can see that where the $B_z$ flux remains ($R < 15R_E$), the current sheet continues to be thick. Combining all the sample fits over several cycles of loading and unloading results in the picture shown in Fig. 7: a dawnward current sheet moving between thick and thin states depending on the level of flux pileup. Indeed, this is a common pattern throughout the simulation: where there is flux pileup, the current sheet is thicker and the current density is lower (e.g. Fig. 9).

This cycle may explain the apparent contradiction between the Liu+ prediction of thinner dawnward current sheets in small magnetospheres and the Poh et al. (2017b) spacecraft observation of thicker dawnward sheets at Mercury. Even though, on average, the current sheet is thinner dawnward (as Liu+ predicts), the sampling of measurements could be producing the



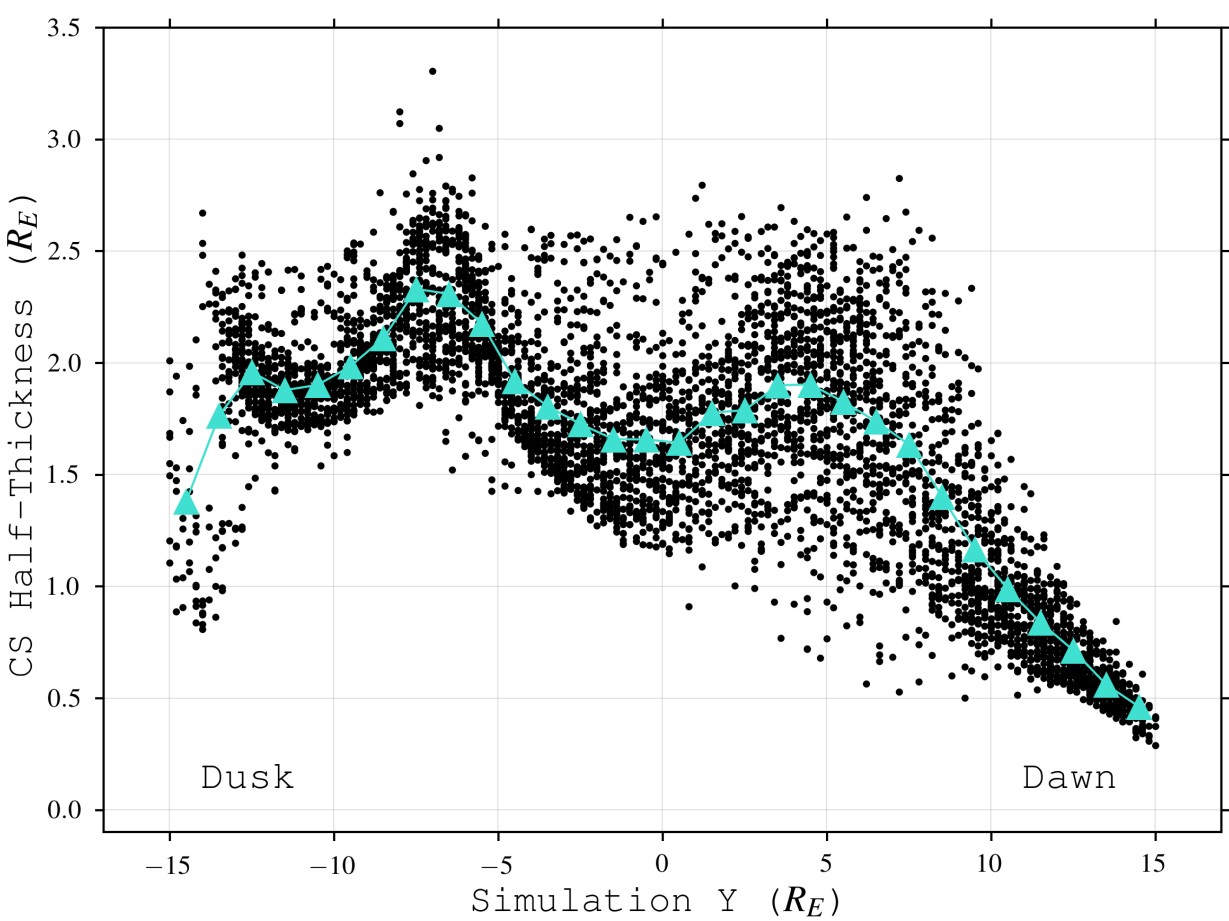

**Figure 7.** Best-fit current sheet half-thicknesses ($L_{CS}$ derived by fitting eq. 9 to 5037 cuts of $B_x$ along the $z$-direction. These cuts were randomly sampled in the tail $xy$-plane and over the simulation time period (see text). There is a bias towards the current sheet being thinner on the dawnside. However, the dawnside also sees a larger spread in thicknesses: this is a result of temporal effects (see main text for discussion).





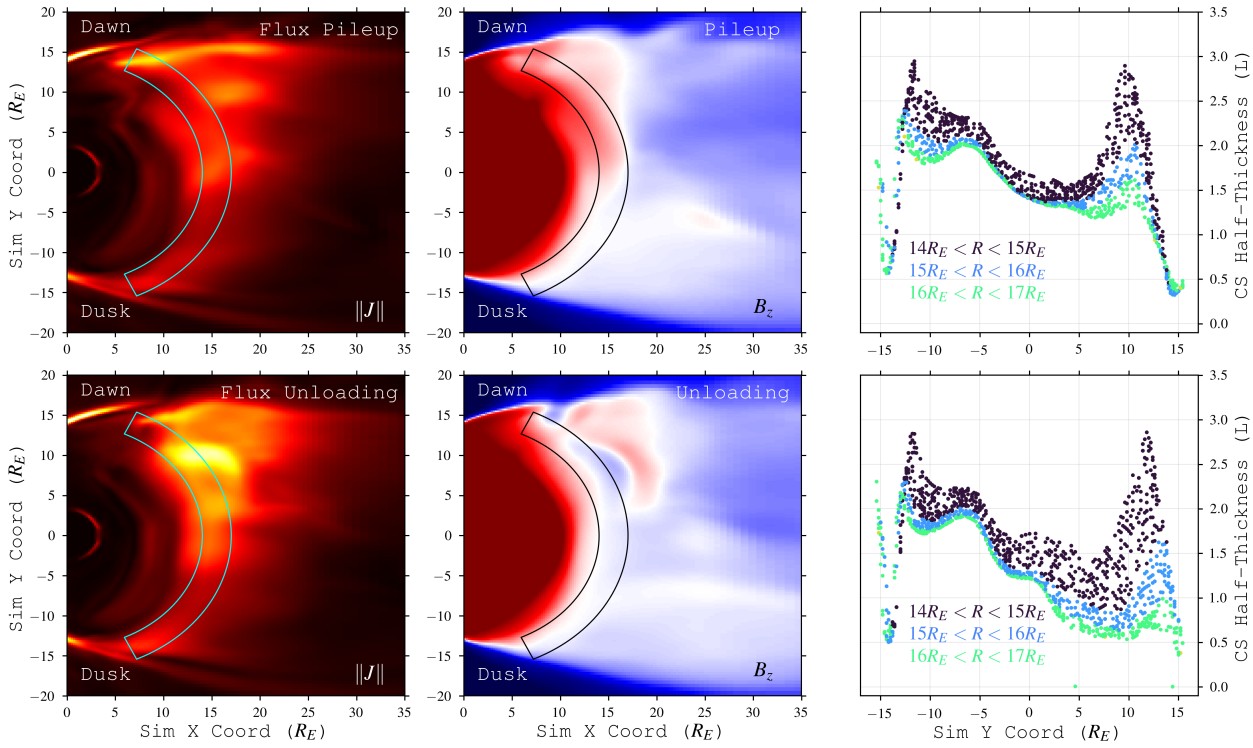

**Figure 8.** Cross-comparison of current sheet density magnitude (left), current sheet $B_z$ flux pileup (center; same parameters as Fig. 4) and sampled thicknesses (right) during (top row) and after (bottom row) a global dipolarization event. Current sheet fits are sampled from the area within the wedges ($14R_E < R < 17R_E$). The current sheet is thick where the $B_z$ flux has piled up, and thin where the flux has been unloaded.

opposite result. As shown in Figs. 7 and 8, the sampled sheet thickness can greatly depend on where and when the craft crosses the tail. In our simulation, the current sheet is continuously morphing between "thick" and "thin" states; both types of regions exist simultaneously within the dawnside. Most points in the tail preferentially see thicker sheets over time, though some preferentially see thinner sheets. It is possible that these effects combine to produce a sampling bias in time and space towards thicker sheets. This will need more investigation, especially with regard to the varying solar wind conditions and seasons that MESSENGER experiences at Mercury.

## 5 Conclusion

We have simulated a small, "mini-Earth" in which the standoff distance and magnetotail width are akin to Earth's as measured in planetary radii, but with the ion-scale length $\delta_i$ set to $1\ R_p$. We find that Hall effects are sufficient to generate tail asymmetries in dipolarization, plasmoids, and current sheet thickness; no electron-scale physics are required, though they may contribute





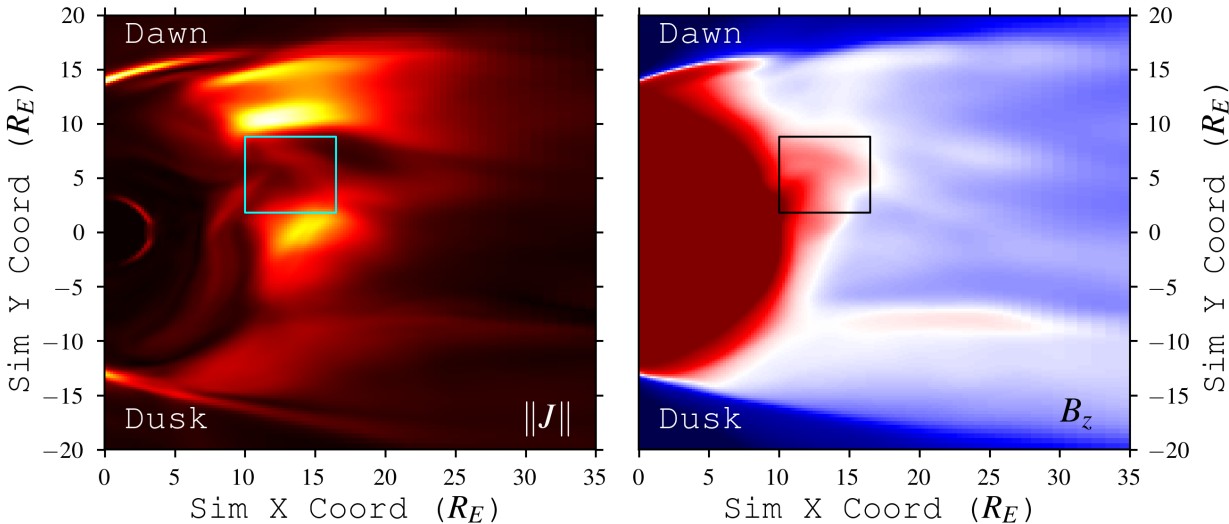

**Figure 9.** Cross-comparison of current density magnitude $\|J\|$ and normal magnetic field $B_z$ in the tail plane at a selected snapshot time. The local pileup of magnetic flux thickens the current sheet, resulting in a lower current density and impeding local reconnection.

to these or other asymmetries. Furthermore, we note that the observed asymmetries in our simulation do not appear in the ideal MHD portion of our run. Thus, we conclude that adding Hall physics is sufficient to generate asymmetry in the tail (in

contrast with Chen et al. (2019), who argue that electron-scale effects are required). However, some questions still remain concerning observed asymmetries at Earth and Mercury and differences between tail asymmetries across system sizes. It is possible that including kinetic effects may better reproduce specific observed asymmetries, though they are not needed for a general explanation of tail asymmetry.

In general, our simulation appears to corroborate the Liu+ picture of tail asymmetry in small magnetospheres; however, the

Lu+ finding that the transported tail $B_z$ thickens the current sheet is readily manifested here. Although the reconnected $B_z$ does drive outflows and thin current sheets on the dawnside, we see that it can pile up and thicken current sheets. There is a continuous cycle between the dawnward transport of $B_z$ leading to pileup (which thickens the current sheet) and reconnection (which thins the current sheet); this manifests in an oscillating current sheet thickness. On average, we find the current sheet is thinner on the dawnside, but it can occasionally be thicker in some regions depending on the level of flux pileup.

Further study will be required to confirm or contrast this picture for larger magnetospheres. Since our simulation is of a "mini-Earth" magnetosphere, several questions concerning more realistic magnetotails remain:

- How does the weaker, offset dipole of Mercury affect the amount of magnetic flux available for transport/pileup and the resulting plasmoid formation/ejection?

- Are the observed dipolarizations at Mercury actually "global", like substorms?





– How does increasing the system size/$\delta_i$ ratio affect tail convection, transport of $B_z$, and plasmoid/DF formation?

    – What other effects (e.g. kinetic, ionosphere) cause asymmetries and how do they interact with one another?

We look forward to future studies which will investigate these questions in greater detail.

*Code availability.* Observational data were not used, nor created for this research; the model algorithm and simulation parameters are described in detail above and in the references.

*Author contributions.* C.B. developed the code, analyzed the simulation results, wrote the manuscript and produced the figures. J.D. edited the manuscript and provided computing resources. Both authors conceptualized the research goals and guided the direction of inquiry.

*Competing interests.* The authors declare that they have no conflict of interest.

*Acknowledgements.* The authors acknowledge funding from NASA's Heliophysics Internal Scientist Funding Model (HISFM) program. C.B.'s software development was partly supported by an appointment to the NASA Postdoctoral Program at NASA-Goddard Space Flight

Center, administered by Universities Space Research Association under contract with NASA. C.B. thanks Alex Glocer for useful discussions concerning simulation techniques. C.B. thanks Ryan Dewey for helpful discussions concerning Mercury tail asymmetries/MESSENGER observations. Observational data were not used, nor created for this research; the model algorithm and simulation parameters are described in detail above.



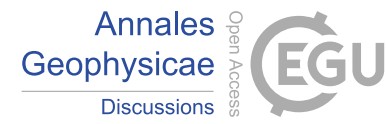

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
