# Peer review of "Magnetotail Reconnection Asymmetries in an Ion-Scale, Earth-Like"

_Annales Geophysicae, 2021_

## Author Response (AR1)

**REFEREE 1**

>The paper describes Hall MHD simulations with an increased inertial length and analyses dawn-dusk asymmetries and temporal variations in the solution.

>Major issues:

>I find the paper's title, abstract and language misleading. While the authors carefully avoid to claim that the modeled system represents Earth, everything implies that to be the intention, including the distance units shown as R_E, comparison of magnetospause stand-off distance etc.

>It would be much better to say that these are Hall MHD simulations of Earth with a drastically (about factor of 70) increased Hall effect, or ions with mass 70amu instead of 1amu.

>Instead of writing the results in normalized units, why not write them out in physical units? With some effort, I managed to figure out that the authors most likely used the following normalization:

>L_0 = 6378km, B_0=10nT, rho_0=5amu/cc, v_0=97.5 km/s, t_0=65s

>This means that the simulations represent the following setup:
>the incoming solar wind velocity is ~400 km/s,
>the solar wind density is 5 amu/cc
>and the IMF strength is 10nT, pointing mostly southward.

>The inner boundary density is set to 20amu/cc.

>These are perfectly fine numbers for Earth (but not for Mercury), nothing unusual about them. The only unusual value is the ion inertial length, which is 1 Re instead of 1/70 Re.  That's OK too as long as it is clearly described. No need to talk about mini-magnetosphere and provide results in normalized (and incomprehensible) units.

>For example the "substorm" frequency is given as 5-10 t_0 (line 187), but with the above constants it actually means 5-10 minutes, which is much easier to interpret.

**RESPONSE:** We do not believe there to be any inconsistency in describing the simulated system as both "small" (relative to di) and "Earth-like" (relative to the magnetopause and bow shock standoff distances). However, we will change our language to use "ion-scale" instead of "small" in order to emphasize that our system size is much closer to di than Earth's would be, even though the rest of the system parameters are representative of Earth.

We note also that there is no ambiguity here with respect to Mercury. Mercury lives in a different magnetospheric parameter space (different dipole strength, different solar wind parameters). However, since both this ion-scale Earth and Mercury share a similar scale size with respect to the ion inertial length, these magnetospheres have important similarities in how the Hall effect impacts plasma behavior and observables. These similarities may shed light on MESSENGER observations.

Yes, these are close to the normalization values we used. We have added the normalizations to the problem setup in order to facilitate discussion and comparison. We will change to use physical units where convenient and appropriate (e.g. "minutes" instead of $t_0$, $R_E$ instead of $L_0$).

A minor correction: since di depends on the square root of the density, the ions in your example would actually be closer to $70^2$ amu. Our selected normalization parameters leads to ions weighing 3942 amu.

>Another major issue is the unnecessarily sharp and sometimes incorrect contrasting with previous work. The manuscript incorrectly claims that MHD-EPIC simulations use ideal MHD coupled with PIC (line 48), when it is clearly stated in the cited papers that the PIC regions are coupled with Hall MHD. Even if Hall MHD was not used in the full domain, it is hard to argue that Hall MHD matters away from current sheets (line 50) in the real systems of Mercury and Earth.

**RESPONSE:** Yes, "ideal MHD" is a bad typo and our mistake. Thank you for catching that. We have clarified that the MHD portion of MHD-EPIC is **Hall** MHD. We have also clarified that it is unclear how the embedding and boundaries of the kinetic regions within Hall MHD affects the local-global feedback dynamics. It may end up not significantly mattering, but we will not know for sure until we are able to compare global magnetospheres from full kinetic, Hall MHD, and coupled kinetic-HMHD simulations.

>While it is stated that a resolution of 20 grid cells per inertial length is needed to get fast reconnection, the manuscript presents simulations that only achieves 5 cells per $d_i$ (line 154), so the criticism of previous work (for example lines 50, 55, 60 etc) with respect to insufficient resolutions seems unfair.

**RESPONSE:** This criticism was meant to apply to our work as well as previous work by other groups, and leads into the following paragraph (L70) which argues that such global models need enormous computing power. GPUs are one possible way to provide this power.

We have modified the introduction to emphasize that *all* current work, including this paper, fails to meet this 20 cell/di limit. We have also clarified that we mean 5 cells/***solar wind*** di; the simulation is able to obtain 10-20 cells/***tail*** di (see new di figure as per minor comment below).

>The paper emphasizes repeatedly that Chen et al 2019 claims that dawn-dusk asymmetry requires electron physics (lines 28 and 245), but in reality that paper compares Hall MHD and MHD-EPIC simulations and finds that both show some asymmetries, but they are not the same. Given the hugely amplified ion-inertial length in this manuscript and the similar grid resolution of only about 5 cells per $d_i$, it is unclear why these results would be more applicable to Mercury than the results by Chen et al 2019, or why the general conclusions found here are better than the conclusions drawn specifically for Mercury by Chen et al 2019.

**RESPONSE:** We wanted to emphasize to the reader that, at minimum, the Hall effect can produce tail asymmetry. We note that Chen+2019, in the last paragraph of Section 4 states:

<<<"In order to demonstrate the importance of including physics beyond Hall-MHD, we compare the MHD-EPIC simulations with pure Hall-MHD simulations. Figure 14 shows the evolution of plasma jets and Bz for Hall-B simulation. This simulation does not show any significant dawn-dusk asymmetry and the results are quite different from those of the MHD-EPIC-B run.">>>

Additionally, Chen+2019, in the conclusion, state:

<<<"There are not any significant dawn-dusk asymmetries of the reconnection products in the Hall-B simulation. In general, Hall-MHD simulations do not appear to match observations very well in terms of dawn-dusk asymmetries of magnetic reconnection. MHD-EPIC simulations contain more physics than the pure Hall-MHD simulations due to the kinetic treatment of both electrons and ions by the PIC code.">>>

Unless we are misunderstanding, these statements by Chen+ seem to imply that kinetic effects are required for dawn-dusk asymmetries, and that Hall MHD is not sufficient. Our results demonstrate that it is possible for Hall MHD to produce *significant* tail asymmetries. However, we acknowledge that we did not simulate the same system as Chen+2019 nor use the same code, so there may be additional nuances that are not being considered. We will be more explicit in stating that our simulations, along with Chen+2019 and Dong+2019, support the idea that tail asymmetry is an universal consequence of the Hall effect in ion-scale magnetospheres (and not specific to magnetospheres that are physically small, like Mercury). We are not superseding any previous studies, but are adding to them.

We also have modified our discussion to make it clear that, although the ion-scale Earth and Mercury simulations have similarities, there may be important differences due to the different physical scale lengths (i.e. bow shock and magnetopause standoff distances). Further studies are needed to understand completely the nuances of how all these length scales interact.

>Minor issues:

>Line 67: while GPUs can help, it is not at all clear how much. How many GPU-s were used? How many CPU-s would achieve similar performance? How long does the simulation take?

**RESPONSE:** For this simulation, we used 8 NVIDIA K20x GPUs. From internal testing of the magnetosphere code, one K20X GPU is comparable to the equivalent of about 70-80 CPU cores. This comparison was made using the same `mpiexec –n 8` command for both GPU and CPU codes: we took the benchmark run time for 8 GPUs and compared it to 8 CPU nodes, resulting in a speedup of about 80x. However, since we did not use an optimized CPU code (e.g. with OpenMP to use multithreading/cores on the CPU nodes), we interpret these results to mean that 1 K20X is roughly equal to 80 CPU cores (not nodes, which may have up to 16 cores each) for this application.

We note that there are several caveats with this particular timing: 1) the CPU code was not optimized; 2) the K20x GPUs used here are now obsolete (newer V100s are about 1.5-2x faster) ; 3) we do not use AMR or other similarly intensive algorithms which use load balancing to redistribute grids/calculations across processors (which would decrease the overall speedup due to GPU memory loading/offloading).

The ideal MHD portion of the simulation took about 2 days to simulate 240 $t\_0$, the Hall MHD portion took about 10-11 days to simulate 57 $t\_0$. The 8 K20x GPUs were able to do about 3.2 steps/second for ideal MHD, and 2.9 steps/second for Hall MHD.

>Line 88: the usual notation is $B\_0$, not $B\_g$. While $B\_0$ is used elsewhere, probably that's the one the authors should rename.

**RESPONSE:** We have changed $B\_g$ to $B\_0$, and changed $B\_0$ to other names where appropriate.

>Also, splitting the magnetic field is not Powell's idea. Tanaka 1994, JCP 111, 381 is a better reference.

**RESPONSE:** We have added the Tanaka reference. Thank you for pointing this out.

>Line 128: Hall MHD has the whistler waves. This needs to be addressed.
How do the authors handle it? What is the whistler wave speed compared
to the fast magnetosonic wave speed? How do they ensure numerical
stability? These issues are extensively discussed in Toth et al 2006.

>The authors should explain how those are addressed by their code.

**RESPONSE:** Since this is a explicit Hall MHD code which uses the Courant condition, we follow Huba 2003 and Toth+2008 in using the whistler wave speed to estimate the maximum wavespeed. We have added this detail to the manuscript.

The steady state timesteps became about 20-25x smaller after switching over to Hall MHD.

>Line 146: fixing the tangential component of $B\_1$ (not B) is somewhat
unusual.

**RESPONSE:** We acknowledge that this is not typically done. As we explained, we had difficulties with density depletion near the inner boundary and experimented with several different combinations of floating/fixing variables at the boundary. The one that worked best to produce a stable evolution without density depletion is presented in the paper.

We have made it more clear that this boundary prescription was arrived at via experimentation, and was driven by numerical considerations rather than physical considerations.

>Line 156: the size of the computational domain should be given (in $R\_E$).

**RESPONSE:** We have added in this information. In normalized lengths, the domain size was $-35.6 < X < 122$ , $-86.4 < Y,Z < 86.4$.

>Equation 9: the offset should not exceed B_0 (to be renamed to B_g).
Is this checked by the fitting script? Should be mentioned.

**RESPONSE:** As long as one side of the current sheet sample has B_x > 0 and the other side has B_x < 0, the best-fit offset will always be less than the best-fit B_0 (which we have renamed to B_a). Indeed, checking the fitting data shows that the offset is less than B_a for each valid fit ( L>0, chi_sq < 0.01).

>Line 264: I don't know what ANGEO policies are, but "code availability" is
not the same as "algorithm described in detail", and in fact, it is not
described in detail.

**RESPONSE:** We will seek clarification on this policy.

>Several figures have no scales: figures 2, 4, 5, 6, 8 and 9. There should
be color bars with physical units.

**RESPONSE:** The plots have been remade to add colorbars, and the normalization values are added to the captions. Other changes have been made in accordance with Referee 2's comments.

>The authors should add an extra figure with the density and inertial
length distribution.  The inertial length in the solar wind seems to
be about 1/12 Re with the above parameters (rho=5amu/cc). Where does the 1/70 come
from?

**RESPONSE:** For the real Earth, di = 101km = 1/70 R_E in the solar wind with the given parameters in the paper. In the tail, due to a lower density, Earth's di is closer to 1/15 R_E. We are using these numbers for comparison with our simulations.

>Typos and minor corrections:

>Line 34: creates *an* electric field

>Line 71: the the

>Line 110: *a* user-set

**RESPONSE:** Thank you for pointing these out. These typos have been fixed.

>Line 111: the calculation of the current density is different from
Toth et al 2008. That difference actually matters for an implicit
solver, and it probably does not matter at all for an explicit solver.
It would be best to delete this sentence.

**RESPONSE:** We have removed this sentence.

>Line 122: the non-conservative form does not lead to a loss of accuracy.
It leads to incorrect jump conditions across shocks. Since the bow
shock of Earth is not magnetically dominated, the error is relatively small.

>Anywhere else, equation 7 is just as accurate as equation 3.

**RESPONSE:** We have modified the discussion accordingly.

>Line 135: 1/60 and 1/70 are not that different. Delete one of the fractions.

**RESPONSE:** Removed.

>Line 141: $B\_g=3000$ $B\_0 = 30,000$ nT is only true on the magnetic
equator at r=1 R_E.

**RESPONSE:** We have now clarified this.

>Line 165...: use physical units, not t_0

**RESPONSE:** we have added physical units wherever appropriate.

>Figure 5 caption: cyan line -> dashed cyan lines

**RESPONSE:** The caption has been changed.
* * *
**REFEREE 2**

This paper presents results from a numerical experiment using Hall MHD model with scaled ion inertial length. The authors argue that Hall MHD is the cause of tail current sheet asymmetry in the scaled magnetosphere.

**Major issues**:

>The "small, Earth-like magnetosphere" in the title is an ambiguous term. Despite the fact that the input parameters, if converted to physical units, represent most likely a Earth-like system, the scaling factor applied together with some other treatments make the outcome more or less similar to Mercury in terms of normalized units especially in the tail. The model-data comparison is also targeted at Mercury but not Earth in later sections.

**RESPONSE A:** Following this comment and referee 1's comments, we have added in the physical value of the normalizations in order to facilitate the discussion. Additionally, we have changed our use of the term "small" to "ion-scale" in order to clarify that we are simulating a magnetosphere whose system size is closer to di than Earth's would be.

When we say our magnetosphere is "Earth-like", we mean that:

1) The subsolar magnetopause stand-off distance is ~10 $R_p$;

2) The subsolar bow shock standoff distance is about 3 $R_p$ further out than the magnetopause stand-off distance.

These relative distances can be controlled, independently of the physical size of the magnetosphere, by setting three dimensionless parameters:

1) $M_A$, the solar wind Alfvenic Mach number

2) beta_sw, the solar wind plasma beta

3) $b = B_{dip}/B_{sw}$, the ratio of dipole field strength at 1 R to the solar wind magnetic field.

When these three dimensionless parameters are set, the subsolar magnetopause standoff distance, in units of $R_p$, is given by: $R_{mp} \sim (beta\_sw/M_A)1/3$ . Setting b then controls the bow shock standoff distance.

The "ion-scale" classification of the experimental magnetosphere is controlled by setting di, the ion inertial length, relative to the planetary radius $R_p$. Setting di does not affect the standoff distances described above; it only controls the global effects of Hall physics. Thus, it is possible to simultaneously have an ion-scale and Earth-like magnetosphere. "Ion-scale" relative to Hall physics, and "Earth-like" according to the magnetosphere aspect ratio.

Note that there is no ambiguity here with respect to Mercury. Mercury lives in a completely different magnetosphere parameter space than Earth, having a much smaller ratio of dipole B strength to solar wind B strength. However, relative to Hall physics, Mercury is much closer to the ion-scale Earth presented in this paper.

The fact that there are similarities between "small Earth" and Mercury does not point to any inconsistency or ambiguity in our approach. It simply means that there are some important similarities between "ion-scale Earth" and Mercury that may shed light on MESSENGER observations. That said, we will modify our discussion to make clear that there may be important differences in "small Earth" vs Mercury dynamics due to the different bow shock and magnetopause stand-off distances. Indeed, the interaction between system size and di will be an important future topic for comparing magnetospheres.
* * *
>From the MESSENGER observational references mentioned in this manuscript, the local PIC simulation [Liu+2019], and the global Hall/MHD-EPIC simulation [Chen+2019], the readers can know that

1. Mercury's tail flux transport events, or dipolarization fronts, favors the dawnside.
2. Mercury's tail current sheet in z is thicker on the dawnside.

>From the local PIC and global Hall/MHD-EPIC simulation results, the readers are aware that

3. The Mercury-like current sheet is thinner on the dawnside near the reconnection region.
4. Mercury's tail current sheet is thicker in the outflow region on the dawnside.
5. Mercury's tail current sheet asymmetry is less obvious in strong IMF driving cases.

>The asymmetry demonstrated in this manuscript shows a thinner current sheet on the dawnside on average, which is consistent with the 31 di length current sheet local PIC run in [Liu+2019] but opposite to the MESSENGER observation at Mercury. In Section 4.3, the authors argue with Figure 7 & 8 that this could be due to temporal sampling effect at different stages of the substorm. The explanation is reasonable within a shell of radius r between 14 to 15 planet radii between certain distances away from the center, but not so obvious in outer regions which are probably closer to the center of X-lines. This may indicate that the thickness of the current sheet is far from uniform, and has a dependence on the relative distance from the reconnection region as well as the driving conditions. The normalized units make it relatively hard to interpret the driving conditions in the simulation, especially when comparing against Mercury or Earth observations.

**RESPONSE B:** We realize that there are several differences between the literature and our results. We will be more explicit in describing how our simulation differs from others, especially in terms of magnetospheric parameters and driving conditions. We note that the effective dissipation scale plays a role in determining the overall shape and thickness of the tail current sheet; this is much more difficult to compare directly between our simulation and MHD-EPIC.

As far as the reason for the discrepancy between the local PIC and global Hall/EPIC simulations (and MESSENGER data), we agree with the referee that there may be spatial as well as temporal structure that is not captured in the MESSENGER analysis. Our suggestion about temporal averaging is just speculation, and we will clarify this in the manuscript.

However, the main points from our analysis remain:  1) Plasmoid formation favors the dawn side, 2) the current sheet is on average thinner on the dawnside but there are periods where it is thicker (depending on the phase of the substorm), 3) all of these effects can be explained by Hall electric fields in a small Earth-like magnetosphere, as argued by Liu et al. [2019].

An important additional fact revealed by our study is that this effect is not specific to Mercury (with its particular magnetopause and bow shock standoff distances) but appears to be a universal consequence of the Hall effect in very different regions of magnetospheric parameter space.

**Minor issues**:

**Introduction & Model Description**

6. It would be better to be consistent in the manuscript when using the text "ion inertial length" or the math symbol di. Define it once in the beginning and use the math symbol thereafter would be nicer.

**RESPONSE:** We have made this change.

7. Regarding GPU: it is unclear what advantages GPUs offer in accelerating the Hall MHD model. It would be more intriguing to briefly mention the strengths compared to CPU computing or shorten the description since this manuscript is aimed at science.

**RESPONSE:** We have added a brief description about why GPUs work well for Hall MHD. Essentially, GPUs take advantage of parallelism in order to have a higher throughput for floating point operations. Finite-volume schemes are massively parallel: the calculation of how a computational cell evolves from t to t+\Delta t is independent of similar calculations for other cells. This makes explicit Hall MHD schemes (such as presented in this paper) quite amenable to GPU acceleration.

8. Line 91: GLM is typically used on a regular grid. If the underlying grid for the field components is staggered then by definition the monopole of B is maintained as long as it is initially zero. If it is the case then it may be worthwhile mentioning that briefly.

**RESPONSE**: We used the same grid for the field components as the other MHD variables. We did not use a staggered grid.

9. Paragraph around Line 125: since the constrained transport scheme is not actually implemented and used in the model, the authors may consider removing this part of the context.

**RESPONSE:** We had intended to mention that CT is another way to handle the divergence constraint, but we will remove this paragraph for clarity.

10. Section 3, problem initialization: as mentioned before, it may be worthwhile to state the key normalized quantities (e.g. Mach number, wave speeds) as well which is better to argue and reproduce the experiments. Alternatively, we can also list physical units, if possible, for better comparison with observations.

**RESPONSE:** We have reframed the simulation setup in order to clarify the key normalization quantities, adding physical units where relevant and appropriate.

11. Line 140: Since this is 3D, the center shall be (0,0,0).

**RESPONSE:** Yes, this is correct. We have fixed this typo.

12. Line 145: inner BC float Bperp, 0 Bpar → what is the physical interpretation/numerical consideration for this magnetic field boundary condition?

**RESPONSE:** This is a numerical consideration. We had difficulties with density depletion near the inner boundary and experimented with several different combinations of floating/fixing variables at the boundary. The one that worked best to produce a stable evolution without density depletion is presented in the paper.

We have made it more clear that this boundary prescription was arrived at via experimentation.

13. Line 150, outer BC: the authors do not mention the size of the simulation domain in terms of normalized distance, which may let readers think that the cuts shown below are from the whole domain slices.

**RESPONSE:** We have added the domain size, which (in normalized lengths) is –35.6 < X < 122 , -86.4 < Y,Z < 86.4.

14. Line 153: due to the fact that this numerical experiment is conducted on an Earth-like magnetosphere with no rotation involved, the meaning of dawn and dusk may be ambiguous to readers unfamiliar with the norms. It would be better to mention that even though there are no dipole tilt or rotation, dawn and dusk are used assuming the sun is rising from the east, etc.

**RESPONSE:** This is a good point. We have modified this discussion accordingly to better define the convention.

15. Paragraph around Line 160: this part argues the usage of 5 against 10 cells per di for sake of computational efficiency. However, the bottleneck is ambiguous. On a rough estimation, the presented simulation size with 18e6 cells in 3D of the Hall MHD model in double precision requires 18e6*8*(12+3) ~ 2GB of memory to store the data, and the runtime memory usage can easily be doubled or tripled. This means that using 10 cells per di requires about an order of magnitude more memory, and 20 cells per di requires about 128GB, which may be the real bottleneck but not speed. If this is true, it would be good to mention it in the text.

**RESPONSE:** This is a good point; however, the Hall MHD explicit time step inversely depends on the square of the grid resolution. Doubling the resolution not only doubles the number of computational cells, but it also (in theory) quadruples the number of timesteps required for the same time period.

We have added discussion about the explicit time step in Hall MHD (also addressing a comment from Referee 1).

16. Line 165: it is unclear why the authors choose to run the simulation first with northward IMF, then southward IMF before turning on the Hall term, while later in around Line 196 stating that

shifting in solar wind IMF is not required to sustain generation of substorms. Also, why does the solar wind magnetic field have a small Bx component?

**RESPONSE:** When starting up the simulation from an initial non-magnetosphere condition, the magnetosphere needs time to settle into a quasi-steady state. We found it more stable to start with a northward IMF and then let the magnetosphere evolve into a southward IMF. Similarly, since it would take too long to run the entire simulation under Hall MHD, we evolve the magnetosphere under ideal MHD before turning the Hall term on. Since our goal is to see how the magnetosphere behaves under a southward IMF with the Hall term, we run the magnetosphere until southward IMF in ideal MHD, and then turn the Hall term on. This is a common procedure for magnetospheric simulations: see, e.g. [Chen+2019; DOI:10.1029/2019JA026840; end of first paragraph in Section 4].

The small Bx component mimics the dipole tilt of the Earth (about 10 degrees).

**Section 4.1**

17. Figure 1: would be better to denote the finest cell region with a box to show the "effective" Hall region, even if the Hall term is presumably added to the whole domain. The colorbar range is saturated on the minimum edge, so it would be better to extend the ranges.

**RESPONSE:** We have adjusted the plot with an extended colorbar range, and added markings to the diagram showing the extent of the high-resolution region. We have also added (per Ref 1's comments) a figure showing the value of di through the tail, demonstrating that we are properly resolving the Hall scale.

18. Figure 2: even though it is mentioned on Line 168 that all the results below come from the simulation after flipping and the Hall term turned on, the left subfigure still shows a snapshot from ideal MHD, which probably comes from a time before the Hall term is turned on. The colorbar is missing, so readers are not sure whether the magnitude of current densities are on the same scale. Since the width of the tail current sheet is mentioned in the caption, it would be better to add notations in the figures to point out the estimated widths.

**RESPONSE:** We have added a colorbar and clarified that Figure 2 is intended in a similar manner as Figure 1: comparison and contrast between ideal and Hall MHD.

Additionally, we meant that the "width" in the x-direction varies across the y-direction of the current sheet (not referring to the thickness variation in the z-direction). We apologize for the typo in the caption to Figure 2; we meant to refer the reader to Figures 6 and 8, which show the current density (and its variation) in the xy-plane. We have clarified both these points in the paper.

**Section 4.2**

19. Figure 3: in the electron subfigure, using yellow streamlines makes it harder to identify the lines from the background colored contour of current densities. I suggest changing the choice of streamline color. Since the finest resolution region only goes up to 15 R in the tail, it is unclear what kind of effect Hall term will have in the further downstream tail region.

**RESPONSE:** The finest resolution actually goes down to 20 R_p in the tail, which covers the current sheet presented in these snapshots. We agree that it is unclear what are the downstream effects, and this will have to be a topic for future studies.

We have changed the streamline color to blue.

20. Line 185: the authors mention "small magnetospheres' in the context'. In principle, Hall effect and reconnection exist in magnetospheres of any size, which lead to depolarization. It would be interesting to point out to what extent drifting electrons contribute significantly to the dipolarization processes with respect to the size?

**RESPONSE:** Yes, the Hall effect and reconnection do exist in all magnetospheres. Reconnection is what generates dipolarizations, but that the Hall effect influences where reconnection/dipolarizations occur.

We have added some discussion to this section about how the size of the magnetosphere relative to the Hall scale (di) may affect the relationship between drifting electrons and dipolarization processes. Specifically, we speculate that making the magnetospheres larger relative to di will cause the drifting electrons to penetrate less deeply dawnward, and move observed dipolarizations closer to the tail center. In other words, this type of asymmetry we observe here may be more pronounced in ion-scale magnetospheres and weaker for system size >>> di.

21. Paragraph around Line 190: from observations and currently available simulations, we know that the frequency of substorm occurrence, or broadly speaking, the process of magnetic flux buildup --- release, varies a lot in magnitudes across Mercury, Ganymede and Earth, etc. In the simulations from this paper, the authors state 7 out of dawnside, 0 out of dockside during the 45 t0 interval. There are several questions regarding this statement:

22. How are the events recorded from the simulation?

**RESPONSE:** The events are recorded via visual observation of the output snapshots (similar to what is presented in Figure 4).

23. Which is a closer analogy for this experiment magnetosphere in terms of substorm frequency in nature? Since it is called Earth-like, one would assume that the substorm frequency would be closer to Earth. Is that the case in the experiment?

**RESPONSE:** We clarify that although the magnetosphere is "Earth-like" in physical size, it is ion-scale relative to di (see response to Major issue above for discussion). The key point we are making is that the interaction between system size and di controls the nature and location of substorms, and not necessarily the physical size.

At Earth, the substorm cycle is roughly one hour and the Alfven time ($t\_A$) is about $R\_E/(100km/s) \sim 63$ seconds. So, a substorm cycle is about 60 $t\_A$.

In our simulation, we set the normalized $t\_0$ to the Alfven time. Since we observed seven events in a 45 $t\_0$ duration, this implies that the experiment magnetosphere substorm cycle is on the order of 5-10 $t\_A$. This is a closer analogy to Mercury with respect to substorm cycles.

24. If it is indeed Earth-like in terms of substorm frequency, then the dawn-dusk asymmetry (which is opposite in Earth and Mercury observations) raises another question: how does the Hall effect influence both the magnetic energy pile-up --- release frequency as well as locations? Does the experiment indicate that asymmetry always comes with higher frequency substorms, or vice versa?

**RESPONSE:** It is not Earth-like in terms of substorm frequency (see response to previous question). We don't believe that this experiment on its own can clarify the relationship between substorm asymmetry and frequency, but additional experiments with system size >>> di and/or different magnetosphere standoff distances should clarify this point.

25. Paragraphs around Line 60, Line 160 and Line 205: these contexts contain discussions about grid resolutions. The authors claim that 20-25 cells/di resolution is required to recover the fast Hall reconnection, while only 5 cells/di is applied in the simulation considering the limitations of resources and model. It may not be necessary to argue about the choice of 5 or 10 cells per di since neither is capable of recovering the fast reconnection.

**RESPONSE**: This is a fair point. We will remove that paragraph since it seems unnecessary.

26. Additionally, the authors acknowledge (around Line 205) that the localized instabilities are missing from the simulation due to the under-resolved resolution. The authors may consider emphasizing if neglecting local tail instabilities has an effect in interpreting simulation results.

**RESPONSE:** Local tail instabilities would produce localized dipolarization fronts, and add more observed events on both dawn and dusk sides of the tail.

However, this would not affect the main result of our analysis: the Hall effect induces dawnward Bz flux transport via electron convection which leads to a dawnward asymmetry in plasmoid/dipolarization formation.

27. Figure 4: colorbar required, maybe with a better choice of color range scale and norm?

**RESPONSE:** We have added the colorbar and slightly widened the color range. However, we did intentionally chose a color range scale to highlight the evolution of the dipolarization event. A more dynamic color range/norm would make it more difficult to see the ejection.

28. Figure 5: Since the middle and right columns show cuts in the xz plane which are different from the left column, it would be very useful to add y axis labels (and ticks). Alternatively, reorganizing the figures such that the xy cuts lie in the first row, while the xz cuts lie in the second and third rows may also work.

**RESPONSE:** We have added y-axis labels to make the shift from xy to xz planes more clear.

**Section 4.3**

29. Line 214: it is mentioned that the sampling is done randomly across the box plane and all times. It is relatively vague about the sampling period (which the reader may assume 45 t0 mentioned earlier) as well as the sampling frequency. Does "random" here indicate uniform sampling?

**RESPONSE:** Thank you for pointing this out, we did mean the 45 t_0 period. "Random" does indicate uniform sampling within this space and across the time period.

30. Figure 7: this plot contains information both in time and space to illustrate the tail current sheet thickness asymmetry. However, temporal effects cannot be directly visualized due to the fact that all sample points are plotted using black dots of the same pattern. One would tend to think that samples from a given snapshot form a continued curved line in the plot. For instance, if one connects the points of the upper and lower envelopes, those shall come from the extreme states in the substorm cycle. The authors may consider adding that kind of information into the figure, using either lines, colors, marker shapes, or sizes.

**RESPONSE:** The current sheet samples are from a 2D area per time dimension, but the plot only shows sampled thickness vs. Y-coordinate. Thus, there may be multiple thicknesses plotted per Y-coordinate (each corresponding to separate X-locations). So, there cannot be a "continued curved line" in the plot; instead, the picture at a given time snapshot looks like what is shown in Figure 8 (the two right-hand plots). There is no "line", but there is a varying spread of thicknesses across the Y-coordinate from dusk to dawn.

We did try a version of this plot with time information denoted by marker color and shape; however, there is significant overlap between times which heavily obscures any relevant information. Thus, we thought it best to separate the overall sampling (Figure 7) from detailed time information (selected sample shown in Figure 8).

The purpose of showing the overall sample (Figure 7) is to demonstrate that random sampling of the current sheet over many positions and times, akin to spacecraft measurements, can lead to a wide range of sampled thicknesses. It is meant to convey two things: 1) the average thickness on the dawnside is less than the average thickness on the duskside; 2) there is a wider variation of thicknesses on the dawnside.

31. Figure 8: between the flux pile-up and flux unloading stages, the dawnside current sheet thicknesses change significantly while the duskside thicknesses are almost constant. Is this the case among the t = 0 ~ 45 t0 simulation time? Does it indicate that there is a strong preference of substorm energy release direction on the dawnside dictated by Hall effect? If so, this could be highlighted in the abstract.

**RESPONSE:** We find that the duskside thicknesses do fluctuate over the 45 t_0 period, but not as dramatically or quickly as the dawnside. The thicknesses generally remain within the range 1.5 R_E – 2.5 R_E, sometimes reaching 3 R_E.

Additionally, the duskside thicknesses have a tighter spread within the sample wedge compared to the dawnside (as seen in Figure 8); this holds over the 45 t_0 period.

We do highlight, in the abstract, the greater frequency of dipolarization events and energy release on the dawnside.

32. Figure 9: a colored contour plot from one snapshot may not be enough to demonstrate the relation between current density and Bz within the current sheet as a function of time. This point may be better explained, e.g., with a 1D line plots of J and B at fixed locations across substorm cycles.

**RESPONSE:** Thank you for the suggestion. We have switched out the plot for the 1D line plots of J and Bz at a fixed location in the tail across the time period, with arrows indicating where pileup of B_z thickens the current sheet.
* * *
**CHANGES MADE TO MANUSCRIPT**:

In the referee responses above, we have noted where we have made changes in accordance with the comments. A summary of the major changes is provided here:
- Added colorbars to several figures
- Added new figure showing density and ion inertial length in the tail
- Removed figure snapshot of Bz/|J| relationship pileup thickening current sheet and replaced it with figure of stationary 1D line cuts over time period showing same effect
- In Methods and Code:
  - Added discussion of explicit Hall MHD timestep
- In Problem Initialization Section:
  - Added discussion of dimensionless magnetospheric parameters
  - Added normalization values
- In Results and Discussion:
  - Changed "small" to "ion-scale" to clarify confusion in comparing between experimental magnetosphere and Earth/Mercury
  - Clarified comparisons with Mercury/Earth in interpretation of results

Other minor changes are noted within the responses to the referees.

---

## Author Response (AR2)

**Referee 1:**

>The revised paper addresses most of my concerns. I still find some of the arguments slightly subjective, but they are not completely wrong. The examined magnetosphere is some hybrid between Earth and Ganymede. It can be studied, of course, but its relevance to any real magnetosphere is unclear. The choice of parameters is guided by the available computational resources and the inefficiency of the numerical method, not by physics. It is not very clear what new insight is gained and how this can be a better approach than using semi-implicit scheme for Hall MHD or embedded PIC, for example. It does not look like GPUs by themselves can overcome the computational challenges of modeling a realistic magnetosphere (Earth or Mercury). Using optimal numerical schemes is more powerful than raw hardware speed.

>In any case, here are a few typos/statements to fix. Line numbers refer to the revised PDF file:
>3: it is unclear what the inflation factor is because there are three d_i values mentioned. It would be best to specify the factore explicitly:
"...artificially inflated by a factor of 70." I also note that d_i in the solar wind is not representative. There are regions with smaller d_i (sheath) and there are regions with larger d_i (most of the tail), as seen in the new Figure 3.

**RESPONSE:** Thank you for pointing out the vagueness. We have modified the abstract to explicitly mention the factor of 70 increase in di.

>5: smaller than Earth's. -->
smaller than Earth's compared to $\delta_i$.
smaller than Earth's in units of $\delta_i$.

**RESPONSE:** We have made this change.

>102: is the electron charge "e" negative? Usually "e" means the elementary charge, and it is positive. But then the sign of the last term in (1) would be wrong. If "e" is indeed negative, this needs to be clearly specified. Maybe q_e is a better notation then.

**RESPONSE:** This was a typo, thank you for catching that. It should have been a (+) sign in front of the Hall term.

>Equations 3-4: Fix the notation: B_g -> B_0

**RESPONSE:** We have made this correction.

>127: across jump shocks -> across shocks across shock jumps (?)

**RESPONSE:** We have made this correction.

>136: Alfven (accent should be \'e not \`e)

**RESPONSE:** We have made this correction.

>150: "cc" is a rather casual notation, and it should be 1/cc anyways.
cm^{-3} is better.

**RESPONSE**: We have made this change.

>150 If L_0=R_E is given in km (not cm), why do you give the v_0 in cm/s?
The usual unit is km/s.

**RESPONSE:** We did the normalization in CGS units, but we had also wanted to use more human-readable units for some quantities (length, mass, etc.). As a compromise, we will give quantities in both the CGS units used for normalization and the more-commonly used units used in magnetosphere studies.

>151: if n_0=5/cc is the number density and rho_0 is the mass density, then
the mean molecular weight \mu should be 1 amu, not 3942.18 amu.

**RESPONSE:** We are not sure what you mean. Rho_0 = n_0 * mass of each ion. In order to have d_i = R_E, then the mass of each ion must be ~3942 amu. We realize that we have used \mu and ion mass interchangably in this paper, so we will change all mentions of \mu to m_i (ion mass) in order to be consistent with the definition of d_i presented in the paper.

>270: an universal -> a universal

**RESPONSE**: We have made this change.

**Referee 2:**

Methods and Codes
Line 91: the abbreviation of GPU has already been mentioned before, so we can use it here. CUDA itself is more of a language, although Nvidia provides multiple low-level libraries written in CUDA.

**RESPONSE:** Yes, CUDA is more like an API than a library. We have made these changes.

Line 150: the authors attempted to use Gauss units based on the choice of G and cm/s, etc. However, in that case why are the lengths still given in km? Maybe it would be more natural here to follow the most common units in magnetophere studies, e.g. km, nT, km/s.

**RESPONSE:** We did the normalization in CGS units, but we had also wanted to use more human-readable units for some quantities (length, mass, etc.). As a compromise, we will give quantities in both the CGS units used for normalization and the more-commonly used units used in magnetosphere studies.

Figure 1: it would be better to add \Delta before y too: Δx, Δy

**RESPONSE**: We have made this change.

Results and Discussion

Line 196: same argument as in Figure 1 caption

**RESPONSE**: We have added \Delta in front of y and z when discussing grid spacing.

Line 203-204: possible typos: based on the colors in Figure 4, ion → green and electron → blue?

**RESPONSE**: Yes, this is a typo. Thank you for catching that.

Figure 6: are the two plots on the left column scaled equally? The gray inner circle seems stretched in the y direction.

**RESPONSE:** We believe that adding the colorbar may have slightly shrunk the x-axis of the plot in the final rendering. Nevertheless, we believe the plot still functions as intended, showing the dipolarization evolution.

Figure 7: same argument as before for the Gauss units.

**RESPONSE**: We have modified the caption to include both Gauss and nT.

Conclusion

Line 284: there is an extra ">" sign.

**RESPONSE**: We have removed this extra sign.
* * *
List of changes in paper:

- All typos mentioned above have been fixed
- One sentence in abstract modified as per referee comments above
- Units in problem setup given in both CGS and other, more human-readable, units